# Proteomic and Global DNA Methylation Modulation in Lipid Metabolism Disorders with a Marine-Derived Bioproduct

**DOI:** 10.3390/biology12060806

**Published:** 2023-06-02

**Authors:** Olaia Martínez-Iglesias, Vinogran Naidoo, Lola Corzo, Iván Carrera, Silvia Seoane, Susana Rodríguez, Margarita Alcaraz, Adriana Muñiz, Natalia Cacabelos, Ramón Cacabelos

**Affiliations:** EuroEspes Biomedical Research Center, International Center of Neuroscience and Genomic Medicine, 15165 Bergondo, Corunna, Spain

**Keywords:** dyslipidemia, cardiovascular disease, DNA methylation, epinutraceutical, epigenetic modulator

## Abstract

**Simple Summary:**

Dyslipidemia is a significant risk factor for cardiovascular disease, and current treatments such as statins can have toxic side effects. RCI-1502, a bioproduct derived from the European *S. pilchardus*, has shown potential as a natural and less toxic alternative treatment for cardiovascular disorders. Alterations in DNA methylation patterns may contribute to the development and progression of lipid metabolism disorders, including atherosclerosis and other cardiovascular diseases. Alterations in DNA methylation may contribute to the development and progression of lipid metabolism disorders, including atherosclerosis and other cardiovascular disorders. In this study, we conducted proteomic analysis of RCI-1502 extracts and classified the identified proteins by their primary molecular functions using the PANTHER classification system. We furthermore investigated the therapeutic potential of RCI-1502 on gene expression and DNA methylation in a high-fat diet mouse model and in patients with lipid metabolism disorders. RCI-1502 treatment regulated the expression of cardiovascular-related genes and reduced DNA methylation levels in both the high-fat diet model and human samples. Furthermore, RCI-1502 regulated cholesterol and triglyceride levels in dyslipidemic patients. Our findings suggest that RCI-1502 is an epigenetic modulator with promising therapeutic potential for dyslipidemia and associated cardiovascular disorders and highlight the potential of epigenetic modulators in treating lipid metabolism disorders. Future studies could explore the combination of RCI-1502 with established treatments for dyslipidemia.

**Abstract:**

Dyslipidemia is a significant risk factor for cardiovascular disease and stroke. Our recent findings showed that RCI-1502, a bioproduct derived from the muscle of the European S. pilchardus, has lipid-lowering effects in the liver and heart in high-fat diet (HFD) fed mice. In the present follow-up study, we investigated the therapeutic potential of RCI-1502 on gene expression and DNA methylation in HFD-fed mice and in patients with dyslipidemia. Using LC-MS/MS, we identified 75 proteins in RCI-1502 that are primarily involved in binding and catalytic activity and which regulate pathways implicated in cardiovascular diseases. In HFD-fed mice, RCI-1502 treatment significantly reduced the expression of cardiovascular disease-related genes, including vascular cell adhesion molecule and angiotensin. RCI-1502 also decreased DNA methylation levels, which were elevated in HFD-fed mice, to levels similar to those in control animals. Furthermore, peripheral blood leukocyte DNA from dyslipidemic patients exhibited higher DNA methylation levels than healthy individuals, suggesting a potential association with cardiovascular risk. Serum analysis also revealed that RCI-1502 treatment regulated cholesterol and triglyceride levels in patients with dyslipidemia. Our findings appear to suggest that RCI-1502 is an epigenetic modulator for the treatment of cardiovascular diseases, specifically in individuals with dyslipidemia.

## 1. Introduction

Dyslipidemia, a metabolic disorder characterized by alterations in the plasma lipid profile, including low-density lipoprotein cholesterol (LDL), triglycerides, and high-density lipoprotein (HDL), is a significant risk factor for cardiovascular disease (CVD) [1]. The causes of dyslipidemia include environmental factors such as diet, tobacco use, and genetic factors [1]. Hypercholesterolemia, characterized by high levels of cholesterol, is the most common form of dyslipidemia and is associated with an increased risk of CVD [2]. In 2019, elevated LDL-cholesterol levels were identified as the eighth leading risk factor for death [2]. Atherogenic dyslipidemia, defined as the coexistence of high triglyceride levels and low HDL levels, is a prevalent trait in individuals with diabetes or metabolic syndrome and has been linked to an increased risk of developing CVD [2].

DNA methylation, a stable epigenetic modification resulting from the addition of a methyl group to DNA, is associated with the pathogenesis of various disorders, including atherosclerosis and CVD [3]. DNA methylation plays a critical role in regulating gene expression by altering the stability of, and accessibility to, DNA [4]. DNA methylation is carried out using DNA methyltransferases (DNMTs), which catalyze the addition of methyl groups to the CpG sites on DNA [5]. DNMT1 maintains inherited DNA methylation patterns [6], and DNMT3a and DNMT3b are responsible for de novo methylation [7]. DNA methylation acts as a repressive mark that recruits other silencing elements, including methyl-CpG-binding proteins [8], and is known to play a crucial role in dyslipidemia and related diseases, particularly atherosclerosis and CVD.

Statins are commonly used to treat hypercholesterolemia. These compounds reduce cholesterol biosynthesis [9] and regulate DNA methylation of several key genes involved in lipid metabolism, including *DHCR24*, *SC4MOL*, *ABCG1* and *NACA* [10]. Similarly, hydralazine lowers high blood pressure and also decreases DNMT expression [11]. SGLT2 inhibitors are used to reduce renal tubular glucose reabsorption, but they also impact DNA methylation and act as epidrugs for treating diabetic cardiomyopathy [11]. Epidrugs target epigenetic alterations that control gene expression, such as DNA methylation and histone modifications, and can thereby alter cellular activities by modifying these epigenetic markers, with potential diagnostic and/or therapeutic benefits.

Biobased natural products used in nutritional interventions are effective in delaying the progression of CVD [12,13]. Natural products are an abundant source of bioactive molecules, including epigenetic regulators. Epinutraceutical bioproducts that use natural substances as active ingredients are an attractive option for therapy since they can be produced using biotechnological procedures that preserve the biological properties of the original species. Most natural molecules are highly complex and can influence more than one epigenetic target, such as DNMTs, micro-RNAs (miRNAs) and histone deacetylases (HDACs). Over the past decades, DNA methylation has been identified as one of the molecular mechanisms that underly the effects of diet on human health [14]. Different dietary factors affect methylation signatures in a range of disorders, including diabetes [15] and obesity [16,17].

LipoEsar is a lipoprotein complex and prototype of a biotechnological product derived from the dorsal muscle of the European *Sardina pilchardus* Walbaum, 1792 [18]. This bioproduct maintains the natural properties of its active molecules. LipoEsar is rich in protein and fatty acids; palmitic acid is the main saturated fatty acid, oleic and palmitoleic acids are the principal monounsaturated fatty acids, and eicosapentaenoic acid (EPA), eicosatetraenoic (ETA) and docosahexaenoic acid (DHA) as the main polyunsaturated fatty acids (PUFAs). LipoEsar is also rich in potassium, phosphorous and calcium, and in vitamins B5 and C [19]. In vitro and in vivo preclinical studies show that LipoEsar lowers serum triglyceride, cholesterol and glucose levels, and modulates immunological parameters by increasing monocyte, leukocyte and lymphocyte cell counts [18,20,21].

Clinically, LipoEsar (750 mg/day for three months) significantly reduces the influence of cardiovascular risk factors in healthy individuals and in patients with chronic hyperlipidemia [18]. LipoEsar reduces elevated total cholesterol and low-density lipoprotein (LDL) cholesterol levels, high blood glucose and uric acid levels, and increases high-density lipoprotein (HDL) cholesterol levels. Furthermore, LipoEsar is hepatoprotective as observed by decreased aspartate transaminase (AST), gamma-glutamyl transpeptidase (GGT), and alanine transaminase (ALT) activity. LipoEsar (1500 mg/day for three months) reduces the size of xanthelasma plaques, a common sign of high blood cholesterol levels, and atherosclerotic plaque size on the aortic wall in subjects with chronic hyperlipidemia [22]. In hyperlipidemic individuals, LipoEsar is effective as a coadjuvant to statin medication, allowing the statin dose to be lowered while avoiding the deleterious effects of statin therapy. An effect similar to that in hyperlipidemic individuals was observed in patients with dementia with concurrent hyperlipidemia, whose medication was supplemented with LipoEsar [22]. Furthermore, healthy participants who received LipoEsar showed increased levels of the lymphocyte markers CD3, CD4, CD25, CD26, CD28, and, CD56, suggesting that LipoEsar stimulates the immune system in healthy individuals [23].

LipoEsar is the structural base of RCI-1502. RCI-1502 has a high protein content and a diverse fatty acid composition, including polyunsaturated (eicosatetraenoic, eicosapentaenoic, and docosahexaenoic acids), monounsaturated (palmitoleic and oleic acids), and saturated (palmitic acid) fatty acids [19]. It also contains the essential vitamins B5 and C, and minerals such as potassium, phosphorus, and calcium. Supplementing high-fat diet (HFD)-fed mice with RCI-1502 reduces atherosclerotic plaque progression and cardiovascular damage, and reduces metabolic disturbances associated with obesity and hypercholesterolemia [19]. Specifically, RCI-1502 modulates LDL, HDL, triglyceride, and hypersensitive-C reactive protein levels, and prevents lipid accumulation. Proteins derived from sardine byproducts are rich in essential amino acids that help promote weight loss, reduce lipid levels, provide antioxidant benefits, potentially prevent atherosclerosis, and mitigate obesity-associated complications [24]. Fish omega-3 PUFAs, including EPA and DHA, have diverse effects on several cardiovascular parameters such as arterial pressure, endothelial cell function, blood lipids and inflammation, which play crucial roles in the development of CVD [25]. Therefore, the high concentration of fish proteins and lipids in RCI-1502 suggests that this bioproduct may be a therapeutic option for individuals with lipid metabolism disorders.

The aims of the present study were to: (i) determine the composition of RCI-1502, a bioproduct derived from the muscle of the European *S. pilchardus*; (ii) evaluate the effects of RCI-1502 on the expression of several CVD-related genes in an HFD murine model; (iii) investigate the effects of RCI-1502 on DNA methylation in an HFD mouse model and in human samples from healthy and dyslipidemic subjects; and (iv) examine the impact of RCI-1502 on triglyceride and cholesterol levels in humans. Our study suggests that RCI-1502 is a promising epigenetic modulator for the treatment of CVD.

## 2. Materials and Methods

### 2.1. Ethical Approval

This study was conducted in compliance with the Helsinki Declaration, Spanish law (Organic Law on Biomedical Research, 14 July 2007) and following the approval of the Ethical/Research Committee of the EuroEspes Biomedical Research Center (Epinutra EE2022).

### 2.2. Human Blood Collection

After obtaining informed patient consent, samples were collected at time zero and one month later. Blood samples were obtained from individuals in the supine position following an overnight fast. After venipuncture, the tubes were allowed to clot at room temperature for 30 min before being centrifuged at 4500 rpm at 4 °C for 10 min to separate the serum from the cell fraction. The serum was aliquoted and stored at −20 °C until analysis. To avoid interday technical variations, biochemical measurements during both visits were performed at the same timepoint. Peripheral blood was collected in EDTA-coated tubes, which were then centrifuged at 3000 rpm at 4 °C for 10 min. The buffy coat was collected and stored at −40 °C for DNA extraction. To study the effects of RCI-1503 treatment on lipid levels, blood samples were collected using dry tubes (BD Vacutainer, Eysins, Vaud. Switzerland) before RCI-1502 treatment (to establish baseline values), and then after one month of treatment with RCI-1502 following a 12 h fast. UV-visible spectrophotometry was used to assess biochemical parameters such as total cholesterol, HDL cholesterol (direct technique), LDL-cholesterol (measured), and triglycerides.

### 2.3. Experimental Animals

Male wild-type C57BL/6 mice (*n* = 12) were bred from established colonies and housed in a controlled environment with a 12 h light/dark cycle and free access to food and water. All experimental protocols were conducted in compliance with the European Community Law (86/609/EEC), the European Directive 2016/63/EU, and the Spanish Royal Decree (R.D. 1201/2005), and were granted approval by the Ethics Committee of the EuroEspes International Center of Neuroscience and Genomic Medicine (EB/2015-037).

### 2.4. Preparation of the Marine Lipoprotein Extract (RCI-1502)

RCI-1502 is a lipoprotein complex obtained from *Sardina pilchardus* using non-denaturing biotechnological methods that preserve the natural properties of its active ingredients. The method used for extraction utilized lyophilization, in which the frozen product is exposed to a vacuum to remove water. During this process, the ice transitions directly from a solid state to a vapor state without passing through a liquid phase. RCI-1502 is protein-rich and contains a variety of fatty acids, including saturated (mainly palmitic acid), monounsaturated (primarily oleic and palmitoleic acids), and polyunsaturated (mainly eicosapentaenoic, eicosatetraenoic, and docosahexaenoic acids) [19]. This bioproduct is also a good source of vitamins (mainly B5 and C) and minerals (primarily calcium, phosphorus, and potassium). To prepare RCI-1502 pellet biscuits, RCI-1502 powdered extract (50%) was combined with diet wheat and Milli-Q water (10%; Millipore, Burlington, MA, USA), and the mixture was dried overnight at 34 °C. The resulting food pellets were then stored in air-tight containers at 4 °C.

### 2.5. HFD-induced Mouse Model of Obesity

To generate a murine HFD-induced model of obesity, corn oil, containing a high proportion of triacylglycerol (99%) and comprising 59% polyunsaturated fatty acids (PUFA), 24% monounsaturated fatty acids, and 13% saturated fatty acids was added to food pellets. To prevent any undesirable effects from rancidity or oxidation of the fatty acids, daily feed exchanges were performed. RCI-1502 was prepared as pellet biscuits by combining RCI-1502 powdered extract (50%) with diet wheat and Milli-Q water (10%; Millipore), and left to dry overnight at 34 °C. The food pellets were stored in air-tight containers at 4 °C. To prevent any undesirable effects from rancidity or oxidation of the fatty acids, daily feed exchanges were performed.

In mice, an HFD causes 50–60% of the total caloric intake to be derived from fat, compared to 10–15% in control animals [19]. Furthermore, C57BL/6 mice fed an HFD comprising nearly 60% of the calories from lipids are prone to obesity and associated metabolic disturbances [26]. In the current study, male mice (*n* = 12; 16 weeks old) were randomly divided into three groups of four animals each and were then administered the following feeding regimen: Group A: normal diet for four weeks; Group B: HFD for four weeks; and Group C: HFD for three weeks, followed by one week of treatment with an HFD in the presence of RCI-1502 (Table 1). More specifically, at the start of week 4, the mice in Group C were switched to an HFD supplemented with RCI-1502 for seven days. Food pellets containing only corn oil were mixed with RCI-1502 (2.53–5.06 mg/day/mouse) and recompacted into pellets. This dosage is roughly equivalent to a human dosage range of 750–1500 mg RCI-1502/day for a person weighing 70 kg. The administered dose was selected to maintain the concentration range in which RCI-1502 may exert beneficial effects against lipid metabolism disorders [18,22].

### 2.6. Collection of Samples from Mice

Tissue and blood samples were collected from mice at the conclusion of the treatment regimen. Blood samples were obtained directly from the ventricle and were centrifuged at 4500 rpm. The serum was then collected and stored at −80 °C. The animals were then perfused transcardially with 0.9% NaCl. The liver was removed and either stored at −80 °C for DNA extraction or preserved in RNAlater solution for RNA extraction (Qiagen, Valencia, CA, USA).

### 2.7. DNA Extraction

The Qiagen DNA Blood MiniKit (Qiagen) was used for DNA extraction, which included an initial phase of erythrocyte hemolysis as per the manufacturer’s protocol. DNA was extracted from mice livers using the Qiagen DNA Mini Kit (Qiagen). An Epoch Microplate Spectrophotometer was used to determine the quality and concentration of DNA. Only DNA samples with 260/280 and 260/230 ratios above 1.8 were used.

### 2.8. RNA Extraction

Total RNA from peripheral blood lymphocytes was extracted using the PureLinkTM RNA Mini Kit (Invitrogen, Waltham, MA, USA). Total RNA from mice livers was obtained with the RNAEasy Mini Kit (Qiagen). Briefly, samples were centrifuged to remove the Qiazol reagent and treated with lysis buffer and 2-mercaptoethanol. The lysates were then transferred to purification columns and exposed to Pure-LinkTM DNAse (Invitrogen). After several washing steps, RNA was eluted with RNAse-free water, and the concentration and purity of RNA was determined. Only RNA samples with 260/280 and 260/230 ratios greater than 1.8 were used in this study.

### 2.9. Quantification of Global DNA Methylation (5 mC)

Global 5 mC levels were measured by an ELISA-like colorimetric methylated DNA quantification kit (EpigenTek, New York, NY, USA). DNA (100 ng) was used, and after several reactions and washes, the absorbance (450 nm) was measured. We used a standard curve and linear regression (Microsoft Excel 14.0) to measure the absolute quantity of methylated DNA. Five mC levels are expressed as the mean (%) ± S.E.M.

The following formulae were used to calculate the percent and amount of 5 mC:5 mC (%) = 5 mC (ng)/DNA sample (ng) × 100
5 mC (ng) = (Sample OD − Blank OD)/(Slope × 2)

### 2.10. Quantitative Real Time RT-PCR

RNA was reverse-transcribed following the specifications of the High Capacity cDNA Reverse Transcription Kit (Applied Biosystems, Waltham, MA, USA). RNA (400 ng) was used for the retrotranscription reaction under the following thermocycling conditions: 10 min at 25 °C, then 120 min at 37 °C, and 5 min at 85 °C.

*APOE*, vascular cell adhesion molecule (*VCAM*), angiotensinogen (*AGT*), Angiotensin Converting Enzyme (*ACE*), *ABCB7*, *DNMT1*, *DNMT3a,* and *HDAC3* expression were quantified by qPCR using the StepOne Plus Real Time PCR system (Applied Biosystems), following the manufacturer’s instructions. The Taqman Gene Expression Master Mix (NZYTech) and TaqMan probes (Thermo Fisher, Waltham, MA, USA) were used for PCR, in duplicate (Table 2). Relative quantification was performed using the comparative CT method [27] with the StepOne Plus Real Time PCR software. Human glyceraldehyde-3-phosphate dehydrogenase (GAPDH) was used to normalize the data. Data are reported as fold change relative to healthy samples and are presented as mean ± S.E.M.

### 2.11. Preparation of RCI-1502 for Liquid Chromatography-Tandem Mass Spectrometry (LC-MS/MS) Analysis

RCI-1502 LC-MS analysis was conducted with a Proteomic Platform (INIBIC, A Coruña, Spain). Protein extractions were performed using 1 mg of lyophilized (RCI-1502) samples diluted in 100 µL lysis buffer (2% SDS/6 M urea/25 mM Ambi). After vortexing and sonication (3 min), the samples were centrifuged at 13,000 rpm to remove cell debris and other insoluble materials. Proteins were precipitated with acetone at −20 °C and resuspended in 6 M urea/2 M thiourea/25 mM Ambi. Protein concentrations (at 590 nm) were estimated with a Bradford assay. The samples were confirmed to be of high quality through SDS-PAGE and silver nitrate staining.

### 2.12. LC-MS/MS Analysis

For LC-MS/MS, an in-solution digestion approach to fragment proteins was used. First, equal amounts (20 µg) of each sample were reduced with 10 mM dithiothreitol at 37 °C for 1 h. Then, the reduced molecules were exposed to 50 mM iodoacetamide for 45 min at room temperature in the dark. Next, sequencing-grade modified trypsin (Promega, Madison, WI, USA) was used to hydrolyze the samples at an enzyme-to-substrate ratio of 1:40, and the enzymatic digestion was performed at 37 °C for 16 h. To acidify the peptides, 10% trifluoroacetic acid was used until the pH was approximately 3. The digested peptides were then extracted using in-house stage tips (3M Empore SPE-C18 disk, 47 mm, Sigma Aldrich, St. Louis, MO, USA), dried under vacuum conditions (Thermo, Waltham, MA, USA), and resuspended in water with 2% acetonitrile (ACN) and 0.1% formic acid (FA) for direct liquid chromatography-mass spectrometry (LC-MS) analysis. LC was performed using a nanoElute instrument (Bruker Daltonics, Hamburg, Germany) coupled with a high-resolution trapped ion mobility spectrometry quadrupole time-of-flight (TIMS-QTOF) mass spectrometer (timsTOF Pro, Bruker Daltonics) with a CaptiveSpray ion source (Bruker Daltonics). Chromatographic separation was conducted at 50 °C with a flow rate of 500 nL/min, using a reverse-phase column (15 cm × 75 m i.d.) filled with 1.9 m C18-coated porous silica beads (Dr. Maisch, Ammerbuch-Entringen, Germany), with a pulled emitter tip. Separation was achieved with a linear gradient of 5–35% buffer B (100% CAN, 0.1% FA) over 40 min. Finally, after electrospray ionization (ESI), the peptides were analyzed in a data-dependent mode with the Parallel Accumulation–Serial Fragmentation (PASEF) enabled.

### 2.13. Proteomic Data Analysis

PEAKS Studio 10.6 build 20201221 (Bioinformatics Solutions Inc., Waterloo, ON, Canada) was used to process the mass spectrometry raw files. The MS/MS spectra were matched to in silico derived fragment mass values of tryptic peptides against the UniProt/Swiss-Prot database for *Actinopterygii*. The search parameters were: fragment mass error tolerance: 0.05 Da; parent mass error tolerance: 15.0 ppm; fixed modifications: carbamidomethylation; enzyme: trypsin; variable modifications: acetylation (N-term), acetylation (Protein N-term), oxidation (M), deamidation (NQ); and max variable PTM per peptide: 3. The spec value, based on peptide spectrum matches, showed the relative abundance of the proteins in each sample. Proteins identified with <2 unique peptides were excluded from the analysis. The bioinformatic PANTHER program was used for protein classification to facilitate the overall analysis.

### 2.14. Statistical Analysis

Proteomic data were analyzed with normalization tools and the statistical software ProteinPilot. The data were then exported to Microsoft Excel for further analysis. The normality and equality of variances were evaluated using the Shapiro–Wilk test and Levene’s test for studies with mice. For mice, statistical significance was determined using a non-parametric Kruskal–Wallis test with post hoc Dunn’s correction for multiple comparisons (SPSS software, IBM, Armonk, NY, USA); the data are presented as median ± interquartile range (IQR). In the case of human studies, the Shapiro–Wilk test and Levene’s test were used to evaluate the normality and equality of variances. Statistical significance was determined with paired t-tests (SPSS). Data are presented as means ± S.E.M; *p* < 0.05 was considered statistically significant.

## 3. Results

### 3.1. Proteomic Profiling of RCI-1502

To evaluate the potential functions of RCI-1502, we first performed, in collaboration with Instituto de Investigación Biomédica de A Coruña (INIBIC, Spain), a comparative proteomic study using LC-MS/MS with whole extracts of RCI-1502. For protein identification, the UniProt/Swiss-Prot database for *Actinopterygii* was used by searching for each MS/MS spectrum. Seventy-five (75) proteins were identified with a false discovery rate of 1%, including two or more peptides with a confidence of at least 95% and a protein pilot total score >2. This list included myosin, tropomyosin, ATP synthase, triosephosphate isomerase, alphaenolase, heat shock cognate 71KDa, sarcoplasmic/endoplasmic reticulum calcium ATPase, nucleoside diphosphate kinase B, calmodulin, cytochrome C, glycerol-3-phosphate dehydrogenase 1-like protein, acetyl-CoA acetyltransferase, and fumarate hydrolase, all of which are explicitly linked to cardiovascular disorders (Table 3).

The proteins identified in RCI-1502 extracts were categorized bioinformatically using network and pathway analyses through the PANTHER classification system. The molecular activities of these proteins are shown using pie charts; each chart represents the molecular and biological functions and signaling pathways of the proteins in the extract. The principal molecular functions of the proteins in RCI-1502 were linked to binding (30.4%) and catalytic activity (47.8%) (Figure 1A). Further analysis revealed proteins involved in cellular processes (41.7%), biological regulation (5.6%), metabolism (30.6%), and responses to stimuli (5.6%) (Figure 1B). In addition, the proteins detected in RCI-1502 appear to regulate multiple pathways, including the tricarboxylic acid (TCA) cycle (8.7%), B and T cell activation (8.6%), cytoskeletal regulation (4.3%), cadherin (4.3%) and heterotrimeric G protein signaling (8.6%), apoptosis (8.7%), and inflammation (4.3%) (Figure 1C).

### 3.2. RCI-1502 Reduces Cardiovascular Disease-Related Gene Expression in an HFD Mouse Model

Our next objective was to investigate the effect of RCI-1502 supplementation on energy metabolism in mice that were fed a saturated HFD. RCI-1502 administration did not significantly impact the body weights of HFD-fed mice, as demonstrated by the absence of any significant increases in body weight among RCI-1502-supplemented HFD-fed mice compared to control or HFD-fed mice at any of the timepoints during the four-week treatment period (Appendix A).

Since cardiovascular disorders involve multiple genes, we then focused on analyzing CVD-related gene expression in liver tissue samples obtained from HFD-fed wild-type mice. The liver plays a key role in regulating lipid metabolism and maintaining systemic lipid homeostasis, and dysfunction in hepatic lipid metabolism contributes to the development of dyslipidemia and CVD [41]. Furthermore, changes in gene expression in the liver can be used as a biomarker for assessing the effects of dietary interventions [42]. One gene of particular interest was apolipoprotein E (*APOE*), which plays a crucial role in cholesterol metabolism and CVD by functioning as a lipid transport protein and a primary ligand for low-density lipoprotein (LDL) receptors [43]. However, *APOE* mRNA levels in the liver were not significantly affected by either the HFD diet or by the intake of RCI-1502 (Figure 2A).

Another gene of interest was *VCAM*, a surface protein that induces monocyte adherence and extravasation to blood vessels and whose expression is elevated in palmitate-treated liver [44]. HFD increased *VCAM* expression four fold in the liver, but RCI-1502 treatment significantly reduced *VCAM* mRNA expression to levels similar to those in control mice (*p* < 0.05) (Figure 2B).

Angiotensin (AGT) is the only precursor of angiotensin peptides, and there is a well-established association between AGT and CVDs [45]. In our study, an HFD caused a 20-fold increase in *AGT* expression in the liver; however, RCI-1502 treatment restored this expression to normal levels (*p* < 0.01) (Figure 2C).

ACE inhibitors are proposed as treatments for CVD [46]. An HFD increased *ACE* mRNA expression over 200 fold, but treatment with RCI-1502 reduced *ACE* mRNA levels to levels similar to those in control mice (*p* < 0.01) (Figure 2D).

In addition, chronic pressure overload causes *ABCB7* deficiency and contributes to a decline in cardiac function [47]. In response to a one-week treatment with RCI-1507, however, *ABCB7* mRNA expression in the liver increased three fold (*p* < 0.05) (Figure 2E). These results suggest that RCI-1502 treatment effectively reduces the expression of genes related to cardiovascular disorders in the HFD mice model and may be therapeutically beneficial for the management of CVDs.

### 3.3. RCI-1502 Reduces DNA Methylation in an HFD Mouse Model

Aberrant DNA methylation has a substantial influence on the transcription and expression of critical regulatory genes, inducing a proatherogenic cellular phenotype, which is the primary driving factor behind the pathological development of cardiovascular disorders [48]. An HFD alters DNA methylation patterns considerably, genome wide [47]. In HFD-fed mice, global DNA methylation levels were three-fold higher than in control animals (Figure 3A). However, RCI-1502 treatment significantly decreased 5 mC levels to values similar to control mice (*p* < 0.05). *DNMT1* mRNA demonstrated a similar pattern, with a more than a 30-fold change in HFD-treated animals; however, this increase was restored to control levels after a one-week treatment with RCI-1502 (Figure 3B).

### 3.4. Patients with Dyslipidemia Exhibit Higher DNA Methylation Levels

Our data in mice suggested that changes in DNA methylation caused by an HFD contribute to cardiovascular pathophysiology. To further assess this association, we used peripheral blood leukocyte DNA from healthy subjects (*n* = 6; age, 54 ± 7.68 years) and patients with dyslipidemia (*n* = 7; age, 58 ± 9.62 years) to analyze the relationship between DNA methylation and cardiovascular risk [49]. Healthy patients that underwent RCI-1502 treatment had no other underlying medical conditions and were not taking any medications. Five mC levels were significantly higher in the dyslipidemic group (3.82 ± 0.21) than in healthy controls (2.9 ± 0.37%) (Figure 4A). Similarly, *DNMT1* mRNA levels increased in patients with dyslipidemia (2.57 ± 0.87%) compared with healthy individuals (1.02 ± 0.2%) (Figure 4B). Our findings suggest that dyslipidemia is associated with higher levels of DNA methylation, which could potentially contribute to increased cardiovascular risk in these patients.

### 3.5. RCI-1502 Reduces DNA Methylation and Regulates Cholesterol and Triglyceride Levels in Patients with Dyslipidemia

We next investigated the effect of RCI-1502 treatment on DNA methylation levels and lipid profiles in 11 male and female patients with (age: 49 ± 5.61 years) and without (age: 39 ± 12.51 years) dyslipidemia (Appendix A). RCI-1502 was administered (per os) to these subjects for one month. Five mC levels decreased from 0.32 ± 0.06% (pre-RCI-1502 treatment) to 0.17 ± 0.02% after treatment with RCI-1502 (Figure 5A). Subsequently, we stratified subjects based on whether they were dyslipidemic. DNA methylation levels decreased significantly only in patients with dyslipidemia (0.34 ± 0.12% pre-RCI-1502; to 0.17 ± 0.03% post-RCI-1502) (*p* < 0.05) (Figure 5C) but not in healthy subjects (0.31 ± 0.07% pre-RCI-1502, to 0.18 ± 0.03% post-RCI-1502) (Figure 5B).

We also examined the effect of RCI-1502 treatment on patient serum lipid profiles, specifically triglyceride levels and the total cholesterol/LDL ratio, before and one month after treatment with RCI-1502. The normal range for the total cholesterol/LDL ratio should be lower than 4.5 in males and 4 in females [50]. There were no significant changes in these parameters in response to RCI-1502 treatment when we analyzed both healthy and dyslipidemia subjects (Figure 5D). However, when we analyzed the subgroup of patients with dyslipidemia, we found that triglyceride levels decreased from 202.75 ± 40.56 mg/dL (pre-RCI-1502) to 164.5 ± 42 mg/dL (post-RCI-1502), but this change was not statistically significant (Figure 5E). Similarly, only in patients with hypercholesterolemia, the total cholesterol/LDL ratio significantly decreased from 4.56 ± 0.23 (pre-RCI-1502 treatment) to 4.22 ± 0.14 (post-RCI-1502) (*p* < 0.05) (Figure 5F,G). These findings suggest that RCI-1502 influences the regulation of lipid metabolism through its ability to lower total cholesterol and LDL-cholesterol levels in patients with dyslipidemia or hypercholesterolemia.

## 4. Discussion

In the present study, we investigated whether RCI-1502, a novel bioproduct derived from the dorsal muscle of *S. pilchardus* Walbaum, 1792, acts as an epigenetic modulator and regulates DNA methylation in lipid metabolism disorders. Our recent findings demonstrated the cardioprotective and lipid-lowering effects of RCI-1502 in a high-fat diet-induced mouse model of obesity [19]. Specifically, RCI-1502 improved antioxidant status, hepatic metabolism, body mass index, and renal function, while moderating major cardiovascular risk markers. Here, we used proteomic analysis to identify the molecular functions of proteins found in RCI-1502 extracts. By employing LC-MS/MS and bioinformatic classification with the PANTHER system, we discovered several protein classes, including binding and catalytic proteins. These findings suggest that RCI-1502 extracts may contain proteins with diverse functions that could potentially contribute to its bioactivity. Analysis of the composition of RCI-1502 uncovered several proteins that are involved in cardiovascular disorders, which may explain the cardiovascular protective properties of this marine-derived bioproduct.

There are positive correlations when comparing global DNA methylation with coronary heart disease [51,52] and acute myocardial infarction [53]. Our current study revealed increased levels in global DNA methylation in blood samples from animals fed an HFD and from patients with dyslipidemia. RCI-1502 treatment reduced global DNA methylation levels, particularly in patients with dyslipidemia, and also reduced triglyceride levels and the total cholesterol/HDL ratio. Proteomic analysis of RCI-1502 revealed the presence of fumarate, which inhibits α-ketoglutarate-dependent dioxygenases that are involved in DNA and histone demethylation [54]. RCI-1502 is rich in omega-3 fatty acids [19]; omega-3 supplementation decreases DNA methylation in blood leukocytes [55].

Statins, a cholesterol-lowering drug, inhibit *DNMT1* expression [56] and DNMT activity [57]. Although statins are generally well tolerated, their use may cause adverse effects such as myopathy, hepatotoxicity, peripheral neuropathy, impaired myocardial contractility, and autoimmune diseases in some patients [58]. Hydralazine, an antihypertensive drug for treating essential or severe hypertension, is a DNMT1 inhibitor [11]. In contrast, RCI-1502 regulates *DNMT1* expression and acts as a natural epinutraceutical bioproduct, suggesting that it acts as a less toxic regulator of DNA methylation. RCI-1502, therefore, appears to be a viable therapeutic option for dyslipidemia and CVD.

ApoE is an apolipoprotein that is involved in regulating lipoprotein metabolism and is important in dyslipidemia and CVD. [43]. It is a component of very-low-density lipoproteins (VLDL), intermediate-density lipoproteins (IDL), LDL, HDL, lipoprotein (a) (Lp(a)), and chylomicrons and regulates their clearance from plasma [59]. Dysfunctional ApoE can cause dyslipidemia and affect the metabolism of triglyceride-rich lipoproteins. The plasma levels of ApoE have been associated with different causes of mortality, including cardiovascular-related death. *APOE* expression is lower in plasma from hypercholesterolemic subjects than in healthy ones [60]. However, there were no changes in *APOE* expression in the livers of mice that were subjected to an HFD. Polymorphisms in the *APOE* gene are associated with CVD, lipid levels, and the lipid-lowering response to statins [60].

In the current study, an HFD increased *VCAM*, *AGT*, and *ACE* mRNA expression, but treatment with RCI-1502 deceased the expression of these markers. Specifically, VCAM is a predictive biomarker in CVD and its inhibition prevents angiotensin II-induced hypertension in mice [61,62]. The VHPK peptide has high affinity for VCAM-1 in atherosclerotic plaques and has been widely used for imaging and targeting of these lesions [63]. To this, a VCAM-1-targeting peptide with potential for treating patients with atherosclerotic plaques has been developed [64]. Furthermore, AGT plays a crucial role in blood pressure regulation [45], and targeting this molecule may represent a novel therapeutic approach for hypertension. The antisense AGT inhibitor IONIS-AGT-L_RX_ significantly reduces plasma AGT levels and is well-tolerated in phase II trials as a monotherapy and as an add on to existing hypertension drugs [65]. One of the main drugs prescribed for hypertension since the early 1980s is the small molecule ACE inhibitor captopril [66]. RCI-1502 treatment reduces *ACE* expression suggesting that it may act as an ACE inhibitor. In our study, *ABCB7* mRNA levels were reduced in the HFD group, and chronic pressure overload caused a decrease in *ABCB7* expression [47]. However, treatment with RCI-1502 increased *ABCB7* transcript levels, suggesting that this bioproduct may preserve cardiovascular function.

The diverse array of essential amino acids in sardine byproducts may explain their antiatherogenic, lipid-lowering, weight-loss, and antioxidant properties [24]. PUFAs such as DHA and EPA found in fish may modulate physiological processes related to lipoproteins and blood lipids and overall cardiovascular function including arterial elasticity and blood pressure. PUFAs may alter lipid metabolism by inhibiting the activity of lipid-synthesizing enzymes such as stearoyl-CoA desaturase-1 and fatty acid synthase [67]. Furthermore, fish omega-3 fatty acids may ameliorate atherosclerosis by reducing monocyte recruitment to aortic lesions [68]. The key components in RCI-1502 are fish lipids and proteins, which have immunoregulatory actions that may contribute to its atheroprotective property. Although the precise mechanisms underpinning the health benefits of RCI-1502 are unknown, its therapeutic effects are most likely because of the combined bioactivity of its active components.

Taken together, our data show that patients with dyslipidemia exhibit increased levels of global DNA methylation. However, the novel epigenetic modulator RCI-1502 effectively counteracted the deleterious effects of an HFD or dyslipidemia on DNA methylation. Notably, RCI-1502 also modulates CVD-related gene expression and the levels of triglycerides and cholesterol, both of which are crucial factors in the pathogenesis of CVD. These data suggest that RCI-1502 may hold therapeutic potential as a preventive or therapeutic intervention for dyslipidemia and associated cardiovascular disorders. Moreover, further studies elucidating the underlying mechanisms of action of RCI-1502, including the tissue-specific effects of treatment with RCI-1502 on gene expression and DNA methylation in different organs, including the heart, may provide additional insights into the potential utility of epigenetic modulators in treating metabolic disorders and CVD.

## 5. Limitations of the Study

To ensure that the mice in our study maintained their energy balance, we provided them with unrestricted access to food. However, we were unable to determine the exact amount of food consumed by each mouse, which may limit the interpretation of our results. Furthermore, we did not assess the palatability of the diets used, which is important for understanding feeding behavior, and particularly in the case of unconventional diets such as with RCI-1502 supplementation. To mitigate any potential effects of rancidity or oxidation, we performed daily feed exchanges, obtained weekly weight measurements, and closely monitored the health and behavior of the mice daily. Our epigenetic studies in mice have shown that using three to five mice per group yields statistically significant results. However, since this may limit the generalizability of our findings, we recognize the importance of using larger sample sizes to confirm and validate our results in future studies. Furthermore, while our ELISA data indicate changes in global DNA methylation after treatment with RCI-1502, it is crucial to recognize that the bioproduct only modifies this specific epigenetic mechanism and that other epigenetic processes may also be involved in the observed effects. Therefore, further sequencing studies are necessary to confirm these findings.

## 6. Conclusions

In conclusion, the present study has demonstrated the potential of RCI-1502, a novel bioproduct derived from the European *S. pilchardus*, as an epigenetic modulator with promising therapeutic potential for dyslipidemia and associated cardiovascular disorders. Our study showed the cardioprotective and lipid-lowering effects of RCI-1502 in an HFD model in mice, which was attributed to its ability to regulate global DNA methylation levels and modulate the expression of genes related to cardiovascular health. Importantly, RCI-1502 was found to be a natural and less toxic regulator of DNA methylation, which may present a safer alternative to currently available medications such as statins. The findings from this study highlight the potential of epigenetic modulators in treating metabolic disorders and CVD and suggest the need for further investigation into the underlying mechanisms of action of RCI-1502. However, future studies could examine the cellular and molecular pathways through which RCI-1502 exerts its effects. To this, it would be interesting to explore the effects of combining RCI-1502 with other established treatments for dyslipidemia and CVD, such as statins, to examine potential synergistic effects. In recent years, the concept of “multi-target epidrugs” has gained traction, suggesting that using a combination of different epidrugs may enhance therapeutic outcomes.

## Figures and Tables

**Figure 1 biology-12-00806-f001:**
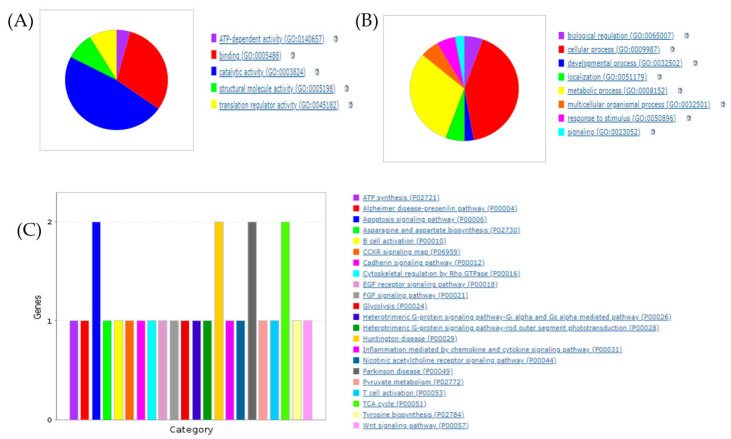
Proteins discovered in RCI-1502, classified according to their molecular functions and biological activities. RCI-1502 extracts (1 mg) were mixed with tagged peptides, fractionated, and separated using nano-flow LC-coupled to a high-resolution TIMS-Q-TOF. This was followed by bioinformatics analysis with the PANTHER classification system. The analysis revealed the molecular functions (**A**), biological processes (**B**), and signaling pathways (**C**) of 75 identified proteins, based on the UniProt/Swiss-Prot databases for ray-finned fish (*Actinopterygii*), which are represented in pie charts. TIMS-Q-TOF, trapped ion mobility spectrometry on a quadrupole time-of-flight.

**Figure 2 biology-12-00806-f002:**
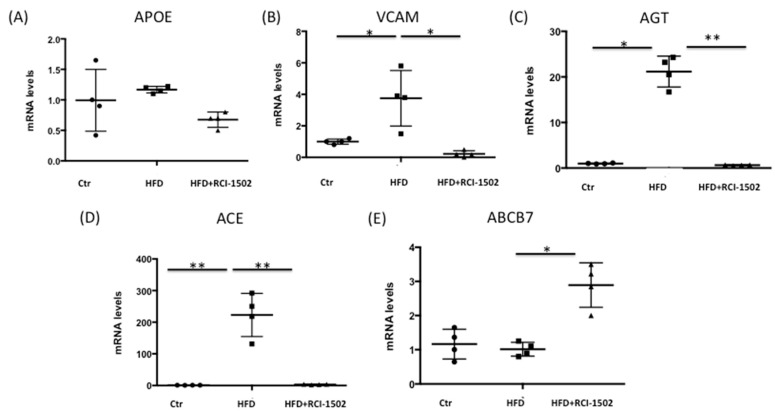
RCI-1502 regulates CVD-related gene expression in mice. *APOE* (**A**), *VCAM* (**B**), *AGT* (**C**), *ACE* (**D**), and *ABCB7* (**E**) mRNA levels were measured using qPCR in liver samples from mice treated with RCI-1502 (*n* = 4) or untreated controls (*n* = 4). Data are expressed as fold change compared to untreated mice and shown as mean ± standard deviation (SD). * *p* < 0.05 and ** *p* < 0.01. CVD, cardiovascular disease; Ctr, control; qPCR, quantitative real time PCR.

**Figure 3 biology-12-00806-f003:**
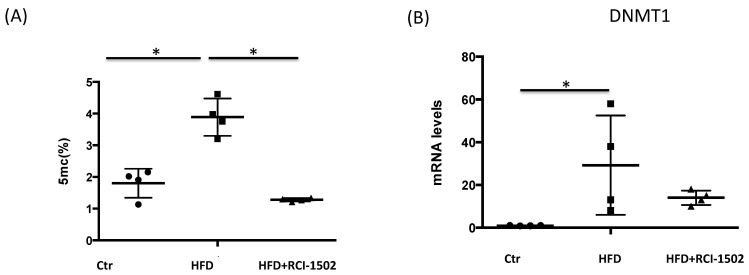
RCI-1502 regulates DNA methylation in mice. (**A**) Global DNA methylation (5 mC) was measured in mouse liver (*n* = 4 per group) and expressed as a percent. (**B**) *DNMT1* mRNA levels were measured using qPCR in liver samples from mice treated with RCI-1502. Data are expressed as fold change compared to untreated mice and shown as mean ± standard deviation (SD). * *p* < 0.05. 5 mC, 5-methylcytosine; Ctr, control; IQR, interquartile range; qPCR, quantitative real time PCR.

**Figure 4 biology-12-00806-f004:**
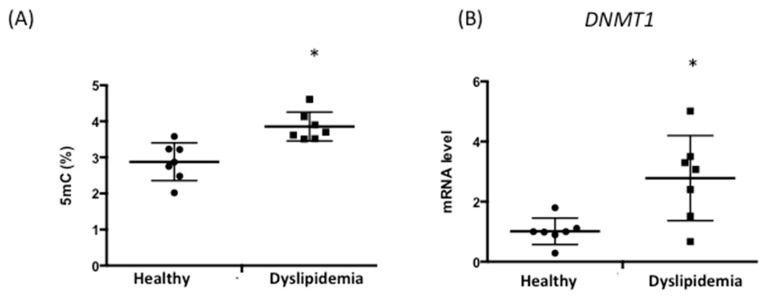
DNA methylation is regulated in patients with dyslipidemia. (**A**) Global DNA methylation (5 mC) levels were measured in buffy coat samples from healthy subjects (*n* = 6) and from patients with dyslipidemia (*n* = 7). (**B**) *DNMT1* mRNA levels were measured using qPCR in buffy coat samples and are expressed as fold induction versus healthy individuals. Data are represented as means ± standard deviation (SD). * *p* < 0.05. 5 mC, 5-methylcytosine. 5 mC, 5-methylcytosine; qPCR, quantitative real time PCR.

**Figure 5 biology-12-00806-f005:**
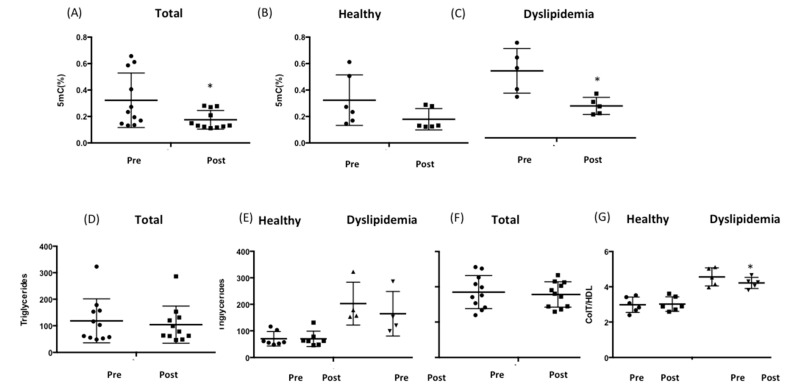
RCI-1502 treatment decreases DNA methylation and total cholesterol/HDL levels in patients with dyslipidemia. (**A**) Global DNA methylation levels were measured in buffy coat samples collected from healthy subjects and from patients with dyslipidemia before (pre-) and one month after (post-) treatment with RCI-1502 (750 mg/day) (*n* = 11). (**B**) Similar to A, but only in healthy subjects (*n* = 6). (**C**) Similar to A, but only in patients with dyslipidemia (*n* = 5). (**D**) Triglyceride levels were measured in serum samples from patients before and one month after treatment with RCI-1502 (*n* = 11). (**E**) Similar to D but differentiating between healthy individuals and patients with dyslipidemia. (**F**) The total cholesterol/HDL ratio was measured in serum samples obtained from patients, before and one month after treatment with RCI-1502 (*n* = 11). (**G**) Similar to F but differentiating between healthy subjects and patients with dyslipidemia. Data are expressed as means ± standard deviation (SD). * *p* < 0.05. 5 mC, 5-methylcytosine. 5 mC, 5-methylcytosine; HDL, high-density lipoprotein; qPCR, quantitative real time PCR.

**Table 1 biology-12-00806-t001:** Mouse diet/RCI-1502 supplementation regimen.

	Week 1	Week 2	Week 3	Week 4
Group A	Normal diet	Normal diet	Normal diet	Normal diet
Group B	HFD	HFD	HFD	HFD
Group C	HFD	HFD	HFD	HFD + RCI-1502

HFD, high-fat diet; RCI-1502, lipoprotein extract from *Sardina pilchardus.*

**Table 2 biology-12-00806-t002:** TaqMan probes.

GENE	ID
VCAM	Mm01320970_m1
ACE	Mm00802048_m1
AGT	Mm005996620_m1
ABCB7	Mm01235358_m1
DNMT1	Mm151854_m1

**Table 3 biology-12-00806-t003:** A selection of cardiovascular-related proteins identified in RCI-1502 extracts.

Protein	Function	Reference
Myosin	Powers heart muscle contraction and increases cardiac function.	[28]
Tropomyosin	Plays a crucial role in cardiac muscle activation; mutations in this protein are associated with cardiomyopathy.	[29]
ATP synthase	Is the primary generator of cellular ATP and is a key regulator of mitochondrial function; ATP synthase dysfunction causes cardiomyopathy and congestive heart failure.	[30]
Triosephosphate isomerase	Deficiency in this glycolytic enzyme causes hemolitic anemia.	[31]
Alpha-enolase	Decreases in the aging heart and may be involved in the cardiomyopathy of aging; improves hypoxia-impaired cardiomyocyte contractility.	[32]
Heat shock cognate 71 KDa	Is a biomarker for poor neurological outcomes in survivors of cardiac arrest.	[33]
Sarcoplasmin/endoplasmic reticulum calcium ATPase	Has decreased expression in congestive heart failure.	[34]
Nucleoside diphosphate kinase B	Deficiency in this protein causes diabetes-like vascular pathology.	[35]
Calmodulin	Calmodulin dysfunction impairs critical cardiac calcium signaling processes.	[36]
Cytochrome C	Myocardial ischemia is related to reduced cytochrome C content.	[37]
Glycerol-3-phosphate dehydrogenase 1-like protein	Regulates cardiac sodium current; a novel mutation in the encoding gene is associated with early repolarization syndrome.	[38]
Acetyl-CoA acetyltransferase	Promotes cardiac repair after myocardial infarction via histone acetylation.	[39]
Fumarate hydrolase	Cardioprotective through the Nrf2 antioxidant pathway.	[40]

## Data Availability

The data presented in this study are available on request from the corresponding author.

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
