# Peer review of "Proteomic and Global DNA Methylation Modulation in Lipid Metabolism Disorders with a Marine-Derived Bioproduct"

_biology, 2023, doi:10.3390/biology12060806_

Round 1
Reviewer 1 Report
The manuscript titled "Epigenetic Modulation of Lipid Metabolism Disorders with a Marine-Derived Bioproduct" investigated the epigenetic regulation in HFD mice models and human patients with dyslepidemia.
Using LC-MS/MS, the authors identified 75 proteins in RCI-1502 that are primarily involved in binding and catalytic activity, and which regulate pathways implicated in cardiovascular diseases. In HFD-fed mice, RCI-1502 treatment significantly reduced the expression of cardiovascular disease-related genes, including vascular cell adhesion molecule and angiotensin. RCI-1502 also decreased DNA methylation levels, which were elevated in HFD-fed mice, to levels similar to those in control animals. Serum analysis also revealed that RCI-1502 treatment regulated cholesterol and triglyceride levels in patients with dyslipidemia. Their study suggest that RCI-1502 is an epigenetic modulator for the treatment of cardiovascular diseases, specifically in individuals with dyslipidemia.
The study looks well performed and written. However a few concerns need to be addressed
Major concern
1) The bioproduct looks promising in terms of the lowering cardiovascualar pathology related genes as well as in dyslipidemia.
In Figure5 B and C, the levels of methylation looks signficantly reduced in RCI-1502 treated healthy individuals. Please justify your findings. Also, there doesn't seem to have a significant difference in methylation levels when comparing a healthy and diseased patient data.
2) Please provide pantient data profiles - age, weight, whether they had other diseases, medication, lipid and glucose values
3) The article from the same group published in IJMS, https://www.medsci.org/v20p0292.html also shows the protective effect of RCI-1502 exhibiting protective CVD and other benefits. Hence in this manuscript, the novel finding here is reduction in DNA methylation. However, the evidences are not convncing enough to prove.
Author Response
Thank you for giving us the opportunity to submit a revised draft of our manuscript titled “Epigenetic Modulation of Lipid Metabolism Disorders with a Marine-Derived Bioproduct”. We are grateful for the helpful criticism given by the reviewers and hope that the revised manuscript will be acceptable for publication as a Research Article in the special issue Nutrient Signaling and Metabolism at the Helm of Cardiometabolic Diseases.
We have incorporated changes that reflect the suggestions provided by the reviewers and have highlighted those changes within the manuscript. The points raised by the reviewers have been addressed as follows (reviewers’ comments are italicized).
REVIEWER 1
Major Comment 1:
The bioproduct looks promising in terms of the lowering cardiovascular pathology related genes as well as in dyslipidemia.
In Figure5 B and C, the levels of methylation looks significantly reduced in RCI-1502 treated healthy individuals. Please justify your findings. Also, there doesn't seem to have a significant difference in methylation levels when comparing a healthy and diseased patient data.
Response:
Thank you to the reviewer. The p-values in Fig. 5A, 5B and 5C are 0.033, 0.11, and 0.041, respectively. Thus, we did not find significant RCI-1502-mediated effects on DNA methylation when we analyzed only healthy patients.
Major Comment 2:
Please provide patient data profiles - age, weight, whether they had other diseases, medication, lipid and glucose values.
Response:
The authors apologize for this omission and thank the reviewer for pointing this out. We have now included, in the revised manuscript, a Supplementary Table 1 (Patient demographics and clinical profiles).
Major Comment 3:
The article from the same group published in IJMS, https://www.medsci.org/v20p0292.html also shows the protective effect of RCI-1502 exhibiting protective CVD and other benefits. Hence in this manuscript, the novel finding here is reduction in DNA methylation. However, the evidences are not convincing enough to prove.
Response:
Thank you for the comment. In that previous study, we showed that RCI-1502 is a potential therapeutic agent for cardiovascular health as it effectively modulates fat-induced inflammation and improves metabolic health. In the current manuscript, our aim was to investigate the underlying mechanisms of the effects of RCI-1502 on lipid metabolism disorders by analyzing its impact on DNA methylation.
In addition to the proteomic analysis of RCI-1502, which showed the presence of various cardiovascular-related proteins, we also found significant reductions in DNA methylation in both animal models and human subjects treated with RCI-1502. These findings provide compelling evidence for the potential therapeutic benefits of RCI-1502 and suggest a possible mechanism of action.
Reviewer 2 Report
1. Please describe details of RCI-1502 in introduction and effects on obesity and cardiovascular diseases?
2. Are RCI-1502 and LipoEsar same or different? Why authors gave more details of LipoEsar?
3. How authors extracted the RC-1502? Give more details of non-denaturing biotechnological process
4. Preparation of RCI-1502 methods copied from author’s previous paper.
5. Remove table 1, it is already published in your recent paper. Cite that paper in methods
6. How human subjects received the RCI-1502 and what is the dose?
7. How much food intake in each group of experimental mice?
8. How about serum lipid profiles of experimental mice after treatment?
9. What are the bodyweights of mice groups before and after the treatment?
10. How authors prepared the RCI-1502 for LC-MS/MS analysis?
11. Fig.1 is not clear, improve the figure 1 resolution?
Author Response
Dr Olaia Martínez-Iglesias
Department of Medical Epigenetics
EuroEspes Biomedical Research Center
Bergondo, 15165
Corunna
Spain
Phone:
E-mail: epigenetica@euroespes.com
1 April 2023
Ms. Shaye Zhang
Assistant Editor
Biology
MDPI
RE: Manuscript ID: biology-2302663
Dear Ms. Zhang,
Thank you for giving us the opportunity to submit a revised draft of our manuscript titled “Epigenetic Modulation of Lipid Metabolism Disorders with a Marine-Derived Bioproduct”. We are grateful for the helpful criticism given by the reviewers and hope that the revised manuscript will be acceptable for publication as a Research Article in the special issue Nutrient Signaling and Metabolism at the Helm of Cardiometabolic Diseases.
We have incorporated changes that reflect the suggestions provided by the reviewers and have highlighted those changes within the manuscript. The points raised by the reviewers have been addressed as follows (reviewers’ comments are italicized).
REVIEWER 2
Comment 1:
Please describe details of RCI-1502 in introduction and effects on obesity and cardiovascular diseases?
Response:
Thank you to the reviewer. We have added information concerning the properties of RCI-1502 and its effects on obesity and cardiovascular diseases, to the Introduction section (page 4) of the revised manuscript.
Comment 2:
Are RCI-1502 and LipoEsar same or different? Why authors gave more details of LipoEsar?
Response:
RCI-1502 is derived from LipoEsar, but it is not the same bioproduct as LipoEsar. RCI-1502 is a modified version of LipoEsar that has a diverse fatty acid composition and a high protein content. We provided more details of LipoEsar because it is the structural base of RCI-1502, a bioproduct that has potential therapeutic effects on cardiovascular disease (CVD). Our intention was to provide information about the natural product (LipoEsar) that was used as a starting point for the development of RCI-1502, including its composition, properties, and preclinical and clinical effects. This information would help us understand the mechanisms of action and potential benefits of RCI-1502, and to compare it with other natural products that have been used in nutritional interventions for CVD.
Currently, the article authored by Carrera et al. (2023) is the only comprehensive source of detailed information on RCI-1502.
Reference:
Carrera I.; Corzo L.; Naidoo V.; Martínez-Iglesias O.; Cacabelos R. Cardiovascular and lipid-lowering effects of a marine lipoprotein extract in a high-fat diet-induced obesity mouse model. Int J Medical Sci 2023, 20, 292-306.
Comment 3:
How authors extracted the RC-1502? Give more details of non-denaturing biotechnological process.
Response:
Thank you to the reviewer. The method for extracting RC-1502 is lyophilization, in which the frozen product is exposed to a vacuum to remove water. During this process, the ice transitions directly from a solid state to a vapor state without passing through a liquid phase. This information has now been included under “Preparation of the Marine Lipoprotein Extract (RCI-1502)” in Materials and Methods.
Comment 4:
Preparation of RCI-1502 methods copied from author’s previous paper.
Response:
We apologize for this oversight. This has now been rectified in the revised manuscript.
Comment 5:
Remove table 1, it is already published in your recent paper. Cite that paper in methods
Response:
Thank you to the reviewer. We have removed this table in the revised manuscript and have added a reference for the paper by Carrera et al. (2023) in our Methods section under “Preparation of the Marine Lipoprotein Extract (RCI-1502)”.
Comment 6:
How human subjects received the RCI-1502 and what is the dose?
Response:
Patients (n = 11) received oral doses (750 mg/day) of RCI-1502.
Comment 7:
How much food intake in each group of experimental mice?
Response:
Since the mice had ad libitum access to food, it was not possible to determine the exact amount of food each mouse consumed. Nevertheless, we have provided the body weights of the groups of mice both before and after treatment with RCI-1502 (new Supplementary Figure S1) (please also see our response to comment 9).
Comment 8:
How about serum lipid profiles of experimental mice after treatment?
Response:
In our previous study (Carrera I et al. 2023., Int J Med Sci.), our group comprehensively characterized the serum lipid profiles in mice fed a high-fat diet (HFD). The total cholesterol levels in the HFD group were 169 ± 5 mg/dL, which decreased to 97.2 ± 4 mg/dL after treatment with RCI-1502. Furthermore, the HDL levels increased from 27.7 ± 5 mg/dL to 57 ± 3 mg/dL, while LDL levels decreased from 13.7 ± 0.3 mg/dL to 9.7 ± 0.2 mg/dL, and triglyceride levels decreased from 100.7 ± 2 mg/dL to 95.5 ± 2 mg/dL.
Reference:
Carrera I.; Corzo L.; Naidoo V.; Martínez-Iglesias O.; Cacabelos R. Cardiovascular and lipid-lowering effects of a marine lipoprotein extract in a high-fat diet-induced obesity mouse model. Int J Medical Sci 2023, 20, 292-306.
Comment 9:
What are the bodyweights of mice groups before and after the treatment?
Response:
Thank you to the reviewer. We have included in the revised manuscript, additional data (Supplementary Figure S1) that show the body weights of the mice in the three treatment groups:
Group A: Normal diet (Control) (n = 4); Group B: High-fat diet (HFD) (n = 4); Group C: HFD supplemented with RCI-1502 (n = 4).
The Kruskal-Wallis tests followed by Dunn’s test was used for statistical analysis, and the threshold of significance was set at p ≤ 0.05.
Comment 10:
How authors prepared the RCI-1502 for LC-MS/MS analysis?
Response:
We have added a description of the preparation of RCI-1502 for LC-MS/MS analysis to the Methods section of the revised manuscript.
Comment 11:
Fig.1 is not clear, improve the figure 1 resolution.
Response:
We have now improved the resolution for Figure 1 in the manuscript.
We sincerely hope that we have addressed the comments to the satisfaction of the reviewers.
Sincerely,
Dr Olaia Martínez-Iglesias
Department of Medical Epigenetics
EuroEspes Biomedical Research Center
Bergondo, 15165
Corunna, Spain
E-mail: epigenetica@euroespes.com
Reviewer 3 Report
The manuscript deals with the epigenetic regulation of lipid metabolism disorders by a marine-derived bio-product (RCI-1502, extracted from the muscle of 135 Sardina pichardus). In the study, the authors use an experimental mouse model fed with different diets and human blood samples from patients with dyslipidemia.
The interest in the identification of epidrugs (which target epigenic alterations), especially if of natural origin, is high in the clinical and therapeutic fields. The topic of the article would then be relevant, however the manuscript has major flaws that need to be fixed.
Specific comments:
1. I think it would be important to know the energy of each diet, which depends both on the relative percentage of nutritional components (lipids, carbohydrates and proteins) and on the quality of sugars and lipids. Furthermore, it would be appropriate to know the dietary intake (and therefore the energy intake) of the animals for each diet. The amount of diet eaten does not depend only on the degree of animal satiety, but also on the diet's palatability. Did the authors explore this issue (this is particularly relevant for not conventional diet receipt, as the RCI-1502 diet is)? This point requires a comment from the authors.
2. In the experimental arm of the study, the sample size is very low and, moreover, how it was calculated was not specified.
3. As a consequence of point (2), given the small number of animals, it would make more sense to use only non-parametric tests and medians (IQR).
4. I'm having trouble framing the experimental design of the study. Replacing HFD (eaten for 3 weeks) with RCI-1502 in the last week and examining the results at the endpoint does not necessarily mean that the readouts is attributable to RCI-1502 treatment. There is much evidence in the literature that discontinuing HFD reverses its effects (especially if administered for a relatively short time, such as 4 weeks). Therefore, in my opinion, it is not clear whether what the authors observe in group C is given by the suspension of the HFD diet or by the assumption of RCI-1502 (or a mix of the two). After all, one of the main features of epigenetic modifications is their dynamic nature and reversibility.
Some additional information could come from an additional group: mice fed for 3 weeks with HFD and with standard diet in the last week.
5. As stated in the text, DNA and RNA are extracted from samples after the whole animal has been perfused with 4% PAF. Although DNA is a more resistant macromolecule, RNA is a very difficult macromolecule to protect during handling of tissue. Although isolation of RNA from formalin-fixed tissue has been reported, the extracted RNA is significantly degraded and can only be used for RT-PCR with amplicon sizes smaller than 200 bp. Aside from degradation of RNA by formalin, there is also structural alteration of template that makes amplification methods poorly reproducible. It is pathologists' dream to have at their disposal a fixative that works histologically and immunohistochemically like PAF and that allows high-quality molecular biology studies to be done from the same samples (and some compounds, alternatives to PAF, begin to approach these performances). I would like a comment from the authors on this point.
6. The study is based on the cardioprotective and corrective effects of RCI-1502 of dyslipidemia and, for the experimental arm, mice fed with HFD are identified as the reference model. However, in the results section, no data identifying the phenotype of the model (e.g., the weight of the animals, their glucose and insulin tolerance, assessment of cardiovascular biomarkers, etc.) is reported.
7. Why study the gene expression of CVD-related genes in the liver? Why not study heart tissue? Can the authors rule out tissue-dependent epigenetic regulations in interpreting and discussing their data? In the literature there is evidence of tissue specificity of nutritional modulation of DNA methylation (e.g., Day J.K. et al, J Nutr. 2002;132:2419S–2423S).
8. Similarly to the experimental arm, in the clinical arm of the study (also suffering of a small sample size), there is no table summarizing patient characteristics, lipid profile, comorbidities, pharmacological treatments and other information useful in phenotyping the recruited patients and the control group for a correct comparison of the results obtained.
9. The sentence "In the present study, we investigated whether RCI-1502.....acts as an epigenetic modulator and regulates DNA methylation in dyslipidemia and cardiovascular disorders" which opens the discussion, in my opinion, is incorrect. In this manuscript there is no evidence of "cardiovascular disorders" neither in the experimental nor in the clinical arm.
10. Finally, in the light of the above comments, due to the potential interest in the proposed bioproduct, the study should be strengthened to support the conclusions drawn by the authors.
Author Response
Dr Olaia Martínez-Iglesias
Department of Medical Epigenetics
EuroEspes Biomedical Research Center
Bergondo, 15165
Corunna
Spain
Phone:
E-mail: epigenetica@euroespes.com
1 April 2023
Ms. Shaye Zhang
Assistant Editor
Biology
MDPI
RE: Manuscript ID: biology-2302663
Dear Ms. Zhang,
Thank you for giving us the opportunity to submit a revised draft of our manuscript titled “Epigenetic Modulation of Lipid Metabolism Disorders with a Marine-Derived Bioproduct”. We are grateful for the helpful criticism given by the reviewers and hope that the revised manuscript will be acceptable for publication as a Research Article in the special issue Nutrient Signaling and Metabolism at the Helm of Cardiometabolic Diseases.
We have incorporated changes that reflect the suggestions provided by the reviewers and have highlighted those changes within the manuscript. The points raised by the reviewers have been addressed as follows (reviewers’ comments are italicized).
REVIEWER 3
Comment 1:
I think it would be important to know the energy of each diet, which depends both on the relative percentage of nutritional components (lipids, carbohydrates and proteins) and on the quality of sugars and lipids. Furthermore, it would be appropriate to know the dietary intake (and therefore the energy intake) of the animals for each diet. The amount of diet eaten does not depend only on the degree of animal satiety, but also on the diet's palatability. Did the authors explore this issue (this is particularly relevant for not conventional diet receipt, as the RCI-1502 diet is)? This point requires a comment from the authors.
Response:
Thank you to the reviewer for this feedback. Regarding the energy of each diet, we have described the nutritional composition of these diets in a previous publication (Carrera I et al., Int J Med Sci, 2023), which reported that the HFD group had a range of 50–60% of kcal from fat and the control group had 10–15%. In addition, a hypercaloric diet with almost 60% of calories from lipids is known to be effective in promoting obesity and metabolic changes (Recena et al., Nutrients, 2019). Please also see our response to Comment 4.
In terms of the dietary intake of the animals, the diets were administered ad libitum, which is a common approach to ensure that the animals consume the amount of food they need to maintain their energy balance. However, we acknowledge that this approach may have limitations, and we did not specifically assess the palatability of the diets in our study. To mitigate any potential effects of rancidity or oxidation, we performed daily feed exchanges, obtained weekly weight measurements, and visually monitored the health and behavior of the mice daily.
While we did not investigate the palatability of the diets, we agree that this is an important factor to consider in understanding the feeding behavior of these mice, especially in the case of unconventional diets such as with RCI-1502 supplementation.
References:
Recena Aydos L, Aparecida do Amaral L, Serafim de Souza R, Jacobowski AC, Freitas dos Santos E, Rodrigues Macedo ML. Nonalcoholic Fatty Liver Disease Induced by High-Fat Diet in C57bl/6 Models. Nutrients. 2019; 11(12):3067.
Carrera I.; Corzo L.; Naidoo V.; Martínez-Iglesias O.; Cacabelos R. Cardiovascular and lipid-lowering effects of a marine lipoprotein extract in a high-fat diet-induced obesity mouse model. Int J Medical Sci 2023, 20, 292-306.
Comment 2:
In the experimental arm of the study, the sample size is very low and, moreover, how it was calculated was not specified.
Response:
We appreciate the reviewer's concern regarding the sample size used in our study. Our Center specializes in nervous system disorders, and we therefore do not have a high number of samples from healthy people or from people with hypercholesterolemia only.
In the current study, we used four animals per group based on our previous experience with mouse models of neurological diseases. We have established that 3–5 mice per group is a good n number for obtaining statistically significant results in our epigenetic studies in mice (Martínez-Iglesias et al., Pharmaceutics, 2022; Martínez-Iglesias et al., Int J Mol Sci, 2022). However, we do acknowledge that having larger sample sizes would be important in future studies to confirm our findings.
References:
Martínez-Iglesias O, Naidoo V, Carrera I, Corzo L, Cacabelos R. Nosustrophine: An Epinutraceutical Bioproduct with Effects on DNA Methylation, Histone Acetylation and Sirtuin Expression in Alzheimer’s Disease. Pharmaceutics. 2022; 14(11):2447.
Martínez-Iglesias O, Naidoo V, Carrera I, Cacabelos R. Epigenetic Studies in the Male APP/BIN1/COPS5 Triple-Transgenic Mouse Model of Alzheimer’s Disease. International Journal of Molecular Sciences. 2022; 23(5):2446.
Comment 3:
As a consequence of point (2), given the small number of animals, it would make more sense to use only non-parametric tests and medians (IQR).
Response:
Thank you to the reviewer for this helpful comment. We have reanalyzed our data in mice (n = 4 per group) using the non-parametric Kruskal-Wallis test with Dunn's post hoc multiple comparisons. These data are presented in the revised manuscript as median ± interquartile range (IQR); the threshold of significance was set at p ≤ 0.05. We have also updated this information under Statistical Analysis in the Materials and Methods section.
Comment 4:
I'm having trouble framing the experimental design of the study. Replacing HFD (eaten for 3 weeks) with RCI-1502 in the last week and examining the results at the endpoint does not necessarily mean that the readouts is attributable to RCI-1502 treatment. There is much evidence in the literature that discontinuing HFD reverses its effects (especially if administered for a relatively short time, such as 4 weeks). Therefore, in my opinion, it is not clear whether what the authors observe in group C is given by the suspension of the HFD diet or by the assumption of RCI-1502 (or a mix of the two). After all, one of the main features of epigenetic modifications is their dynamic nature and reversibility.
Some additional information could come from an additional group: mice fed for 3 weeks with HFD and with standard diet in the last week.
Response:
We wish to clarify, and correct, the experimental design of our study. For the normal diet/HFD/RCI-1502-supplementation regimen, there were three groups of mice:
Group A: Normal diet (Control) (n = 4),
Group B: High-fat diet (HFD) (n = 4),
Group C: HFD supplemented with RCI-1502 (n = 4).
At the start of week 4, the mice in Group C were switched to an HFD supplemented with RCI-1502 (60–75% protein; 15–30% fat; 0–3% carbohydrate; and vitamins and minerals) for 7 days. More specifically, food pellets containing only corn-oil were mixed with RCI-1502 (2.53–5.06 mg/day/mouse), and recompacted into pellets. This dosage is roughly equivalent to a human dosage range of 750–1500 mg RCI-1502/day for a person weighing 70 kg. The administered dose was selected to maintain the concentration range in which RCI-1502 may exert beneficial effects against lipid metabolism disorders (Cacabelos et al., 2005; Lombardi et al., 1999). The bodyweights of the mice in each group were recorded at the end of each week. This information has been included in our description of the HFD-induced mouse model of obesity, and in Table 2, in the Materials and Methods section. We apologize to the reviewer for this oversight.
References:
Lombardi V.R.M.; Cacabelos R. E-SAR-94010: A marine fish extract obtained by advanced biotechnological methods. Drugs Future 1999, 24, 167–176.
Cacabelos R. Pharmacogenomics, nutrigenomics and therapeutic optimization in Alzheimer’s disease. Aging Health 2005, 1, 303–348.
Comment 5:
As stated in the text, DNA and RNA are extracted from samples after the whole animal has been perfused with 4% PAF. Although DNA is a more resistant macromolecule, RNA is a very difficult macromolecule to protect during handling of tissue. Although isolation of RNA from formalin-fixed tissue has been reported, the extracted RNA is significantly degraded and can only be used for RT-PCR with amplicon sizes smaller than 200 bp. Aside from degradation of RNA by formalin, there is also structural alteration of template that makes amplification methods poorly reproducible. It is pathologists' dream to have at their disposal a fixative that works histologically and immunohistochemically like PAF and that allows high-quality molecular biology studies to be done from the same samples (and some compounds, alternatives to PAF, begin to approach these performances). I would like a comment from the authors on this point.
Response:
Thank you to the reviewer for pointing this out. We would like to rectify our earlier statement regarding the samples collected for DNA and RNA analysis. We wish to confirm that we did not use PFA for perfusion of the mice, but instead used 0.9% NaCl. The liver was then immediately removed after perfusion and either stored at -80 °C for DNA extraction or preserved in RNAlater solution for RNA extraction. This information has now been corrected under Collection of Samples from Mice in the Materials and Methods section in the revised manuscript. We apologize to the reviewer for this error.
Comment 6:
The study is based on the cardioprotective and corrective effects of RCI-1502 of dyslipidemia and, for the experimental arm, mice fed with HFD are identified as the reference model. However, in the results section, no data identifying the phenotype of the model (e.g., the weight of the animals, their glucose and insulin tolerance, assessment of cardiovascular biomarkers, etc.) is reported.
Response:
Thank you to the reviewer. We have included in the revised manuscript, additional data (Supplementary Figure S1) showing the weight of the mice in the three treatment groups as described in our response to Comment 4: (Group A: Normal diet (Control) (n = 4); Group B: High-fat diet (HFD) (n = 4); Group C: HFD supplemented with RCI-1502 (n = 4). The Kruskal-Wallis tests followed by Dunn’s test was used for statistical analysis, and the threshold of significance was set at p ≤ 0.05.
In our previous study using a high-fat diet (HFD)-fed mouse model (Carrera I et al. 2023., Int J Med Sci.), we showed that RCI-1502 may serve as a lipo-cardioprotective agent in reducing metabolic disturbances associated with obesity, diabetes, chronic kidney disease, and hypercholesterolemia. In that study, we comprehensively characterized the serum lipid profiles in those mice. The total cholesterol levels in the HFD group were 169 ± 5 mg/dL, which decreased to 97.2 ± 4 mg/dL in response to treatment with RCI-1502. Furthermore, in the presence of RCI-1502, HDL levels increased from 27.7 ± 5 mg/dL to 57 ± 3 mg/dL, LDL levels decreased from 13.7 ± 0.3 mg/dL to 9.7 ± 0.2 mg/dL, and triglyceride levels decreased from 100.7 ± 2 mg/dL to 95.5 ± 2 mg/dL.
Reference:
Carrera I.; Corzo L.; Naidoo V.; Martínez-Iglesias O.; Cacabelos R. Cardiovascular and lipid-lowering effects of a marine lipoprotein extract in a high-fat diet-induced obesity mouse model. Int J Medical Sci 2023, 20, 292-306.
Comment 7:
Why study the gene expression of CVD-related genes in the liver? Why not study heart tissue? Can the authors rule out tissue-dependent epigenetic regulations in interpreting and discussing their data? In the literature there is evidence of tissue specificity of nutritional modulation of DNA methylation (e.g., Day J.K. et al, J Nutr. 2002;132:2419S–2423S).
Response:
Thank you to the reviewer. We chose to study the expression of cardiovascular disease (CVD)-related genes in the liver because this organ plays a key role in regulating lipid metabolism and maintaining systemic lipid homeostasis, and it is known that dysfunction in hepatic lipid metabolism contributes to the development of dyslipidemia and CVD (Alves-Bezerra and Cohen, 2017). Furthermore, changes in gene expression in the liver can be used as a biomarker for assessing the effects of dietary interventions (Mardinoglu A et al., 2018). We have added this information to RCI-1502 Reduces Cardiovascular-Related Gene Expression in an HFD Mouse Model in the Results section of the revised manuscript.
In a previous study using a similar HFD mouse model, RCI-1502 treatment improved hepatic injury markers, as shown by a decrease in GOT (U/L) levels from 264 ± 5 to 162.5 ± 4 and a decrease in GPT (U/L) values from 38.2 ± 3 to 25.7 ± 3 (Carrera I et al., 2023). In addition, there is evidence of a strong correlation between heart disease and liver disease (El Hadi et al., 2020; Naschitz et al., 2000).
We acknowledge the possibility of tissue-specific epigenetic regulation and recognize the importance of careful interpretation of our data. To this, further studies are needed to investigate the tissue-specific effects of treatment with RCI-1502 on gene expression and DNA methylation in different organs, including the heart, to provide a comprehensive understanding of its mechanisms of action and potential as a therapeutic agent for lipid metabolism disorders. This information has been added to the end of Discussion section in the revised manuscript.
References:
Carrera I.; Corzo L.; Naidoo V.; Martínez-Iglesias O.; Cacabelos R. Cardiovascular and lipid-lowering effects of a marine lipoprotein extract in a high-fat diet-induced obesity mouse model. Int J Medical Sci 2023, 20, 292-306.
El Hadi H, Di Vincenzo A, Vettor R, Rossato M. Relationship between Heart Disease and Liver Disease: A Two-Way Street. Cells. 2020, 9(3):567.
Naschitz, J. E., Slobodin, G., Lewis, R. J., Zuckerman, E., & Yeshurun, D. Heart diseases affecting the liver and liver diseases affecting the heart. American Heart Journal. 2000, 140(1), 111–120.
Mardinoglu, A.; Wu, H.; Bjornson, E.; Zhang, C.; et al. An Integrated Understanding of the Rapid Metabolic Benefits of a Carbohydrate-Restricted Diet on Hepatic Steatosis in Humans. Cell Metab. 2018, 27, 559–571.e5
Alves-Bezerra M, Cohen DE. Triglyceride Metabolism in the Liver. Compr Physiol. 2017, 12;8(1):1-8.
Comment 8:
Similarly to the experimental arm, in the clinical arm of the study (also suffering of a small sample size), there is no table summarizing patient characteristics, lipid profile, comorbidities, pharmacological treatments and other information useful in phenotyping the recruited patients and the control group for a correct comparison of the results obtained.
Response:
We have included in the revised manuscript a Supplementary Table 1 that summarizes the patient demographics and their clinical profiles.
Comment 9:
The sentence "In the present study, we investigated whether RCI-1502 acts as an epigenetic modulator and regulates DNA methylation in dyslipidemia and cardiovascular disorders" which opens the discussion, in my opinion, is incorrect. In this manuscript there is no evidence of "cardiovascular disorders" neither in the experimental nor in the clinical arm.
Response:
Thank you to the reviewer. We corrected this sentence in the revised manuscript to now read “In the present study, we investigated whether RCI-1502, a novel bioproduct derived from the dorsal muscle of S. pilchardus Walbaum, 1792, acts as an epigenetic modulator and regulates DNA methylation in lipid metabolism disorders.
Comment 10:
Finally, in the light of the above comments, due to the potential interest in the proposed bioproduct, the study should be strengthened to support the conclusions drawn by the authors.
Response:
We appreciate the reviewer's suggestion to strengthen our study and have carefully analyzed and discussed our findings in the revised version of the manuscript. Based on the comments from all three reviewers, we have incorporated additional experimental data and information to support our conclusions.
In our study, we have provided LC-MS/MS data identifying 75 proteins in RCI-1502 that are involved in binding and catalytic activity, and which regulate pathways implicated in cardiovascular diseases. We have also presented results from our experiments in both a murine model and in human patients with dyslipidemia, demonstrating that RCI-1502 treatment significantly reduced the expression of cardiovascular disease-related genes and decreased DNA methylation levels, which were elevated in HFD-fed mice, to levels similar to those in control animals. Furthermore, we observed that RCI-1502 treatment regulated cholesterol and triglyceride levels in patients with dyslipidemia, suggesting its therapeutic potential for the treatment of lipid-metabolism disorders.
We believe that our findings provide compelling evidence for the therapeutic potential of RCI-1502 for the treatment of lipid metabolism disorders and hope that our study will serve as a basis for further investigations into the potential benefits of RCI-1502 as a therapeutic agent for dyslipidemia.
We sincerely hope that we have addressed the comments to the satisfaction of the reviewers.
Sincerely,
Dr Olaia Martínez-Iglesias
Department of Medical Epigenetics
EuroEspes Biomedical Research Center
Bergondo, 15165
Corunna, Spain
E-mail: epigenetica@euroespes.com
Round 2
Reviewer 1 Report
The authors did respond well to the comments raised. However, in the suplementary table, they mention "patients before- (pre-) and one-month after (post-), treatment with RCI-1502 (750 mg/day)", but only one data point is noted in the table. Please edit it accordingly.
Author Response
Dr Olaia Martínez-Iglesias
Department of Medical Epigenetics
EuroEspes Biomedical Research Center
Bergondo, 15165
Corunna
Spain
Phone:
E-mail: epigenetica@euroespes.com
11 April 2023
Ms. Shaye Zhang
Assistant Editor
Biology
MDPI
RE: Manuscript ID: biology-2302663
Dear Ms. Zhang,
Thank you for giving us the opportunity to submit a revised draft of our manuscript titled “Epigenetic Modulation of Lipid Metabolism Disorders with a Marine-Derived Bioproduct” in response to the comments made by Reviewers 1 and 3. We hope that the revised manuscript will be acceptable for publication as a Research Article in the special issue Nutrient Signaling and Metabolism at the Helm of Cardiometabolic Diseases.
We have incorporated changes that reflect the suggestions provided by the reviewers and have highlighted those changes within the manuscript. The points raised by the reviewers have been addressed as follows (reviewers’ comments are italicized).
REVIEWER 1
Comment:
The authors did respond well to the comments raised. However, in the suplementary table, they mention "patients before- (pre-) and one-month after (post-), treatment with RCI-1502 (750 mg/day)", but only one data point is noted in the table. Please edit it accordingly.
Response:
The authors apologize for this error. We have edited Supplementary Table S1 in the revised manuscript. This table now includes patient demographics and clinical profiles from healthy patients, before and after treatment with RCI-1502.
Reviewer 3 Report
The manuscript is the revised version of a previously submitted one. The authors provided responses to comments made by reviewers. As regards the criticisms raised by me, the answers only partially clarify the doubts.
1. Although the authors corrected the diet of the 3 experimental groups (Table 1), in my opinion the 3 final conditions do not fully correspond to the correct experimental design. As stated by the authors, in fact group B in the last week continued with a diet with 60% energy deriving from fats, while group B went from a diet with 60% energy deriving from fats to one with 15-30% energy from fat, high in protein and low in carbohydrates. My initial criticism remains about the origin of the effects observed by the authors.
2. I appreciate that the authors accepted the suggestion to express the experimental data as median and interquartile range due to the small sample size. However, the conventional way to plot median and interquartile range is the boxplot. Boxplots display minimum and maximum values, the median value, and the lower and upper quartiles.
3. In the penultimate sentence, newly added in the most recent version, the statement "RCI-1502 may also serve as a lipo-cardioprotective agent in reducing metabolic disturbances associated with obesity, diabetes, chronic kidney disease, and hypercholesterolemia." needs to be supported by citations from the literature.
4. I couldn't find Supplementary Figure S1. Also in Supplementary Table T1 the title does not match what is shown. In fact, only one series of data appears for patients and it is not clear whether they refer to pre- or post-treatment with RCI-1502, how they are divided by type and sex, etc.
Author Response
Dr Olaia Martínez-Iglesias
Department of Medical Epigenetics
EuroEspes Biomedical Research Center
Bergondo, 15165
Corunna
Spain
Phone:
E-mail: epigenetica@euroespes.com
11 April 2023
Ms. Shaye Zhang
Assistant Editor
Biology
MDPI
RE: Manuscript ID: biology-2302663
Dear Ms. Zhang,
Thank you for giving us the opportunity to submit a revised draft of our manuscript titled “Epigenetic Modulation of Lipid Metabolism Disorders with a Marine-Derived Bioproduct” in response to the comments made by Reviewers 1 and 3. We hope that the revised manuscript will be acceptable for publication as a Research Article in the special issue Nutrient Signaling and Metabolism at the Helm of Cardiometabolic Diseases.
We have incorporated changes that reflect the suggestions provided by the reviewers and have highlighted those changes within the manuscript. The points raised by the reviewers have been addressed as follows (reviewers’ comments are italicized).
REVIEWER 3
Comment 1:
Although the authors corrected the diet of the 3 experimental groups (Table 1), in my opinion the 3 final conditions do not fully correspond to the correct experimental design. As stated by the authors, in fact group B in the last week continued with a diet with 60% energy deriving from fats, while group B went from a diet with 60% energy deriving from fats to one with 15-30% energy from fat, high in protein and low in carbohydrates. My initial criticism remains about the origin of the effects observed by the authors.
Response:
Thank you to the reviewer. We wish to clarify our previous statement regarding the diets of Groups B and C. The mice in Group B continued to receive an HFD diet with 60% energy derived from fats, which is consistent with our previous publication (Carrera I et al., Int J Med Sci, 2023). In contrast, the mice in Group C were fed the same HFD diet for the first three weeks of the study, but with the addition of RCI-1502 (2.53-5.06 mg/day/mouse) in their HFD diet (60% energy derived from fats) for the remaining one week. The dosage we had used in our study corresponds to a human equivalent of 750-1500 mg RCI-1502/day for a person weighing 70 kg. This information has been corrected under “HFD-induced mouse model of obesity” in the Materials and Methods section of the revised manuscript.
Reference:
Carrera I.; Corzo L.; Naidoo V.; Martínez-Iglesias O.; Cacabelos R. Cardiovascular and lipid-lowering effects of a marine lipoprotein extract in a high-fat diet-induced obesity mouse model. Int J Medical Sci 2023, 20, 292-306.
Comment 2:
I appreciate that the authors accepted the suggestion to express the experimental data as median and interquartile range due to the small sample size. However, the conventional way to plot median and interquartile range is the boxplot. Boxplots display minimum and maximum values, the median value, and the lower and upper quartiles.
Response:
Thank you to the reviewer for this feedback. In the revised manuscript we have now used boxplots to display the median and interquartile range of our experimental data. We have updated Figure 2 and Figure 3 to show the data in this format.
Comment 3:
In the penultimate sentence, newly added in the most recent version, the statement "RCI-1502 may also serve as a lipo-cardioprotective agent in reducing metabolic disturbances associated with obesity, diabetes, chronic kidney disease, and hypercholesterolemia." needs to be supported by citations from the literature.
Response:
The corrected sentence has been integrated into the penultimate paragraph of the Introduction, which now states that supplementing high-fat diet (HFD)-fed mice with RCI-1502 may reduce the progression of atherosclerotic plaques and cardiovascular damage, while also reducing metabolic disturbances linked to obesity and hypercholesterolemia. We have included a reference to support this claim.
However, we have also expanded on the information in that paragraph, where we discuss the relevance of the components in RCI-1502 in the context of lipid metabolism disorders and cardiovascular disease; this information is supported by citations.
Comment 4:
I couldn't find Supplementary Figure S1. Also in Supplementary Table T1 the title does not match what is shown. In fact, only one series of data appears for patients and it is not clear whether they refer to pre- or post-treatment with RCI-1502, how they are divided by type and sex, etc.
Response:
We apologize to the reviewer. We have now provided Supplementary Figure S1 as a separate attachment in our revised submission.
Furthermore, we have edited Supplementary Table S1 (also attached separately) in the corrected manuscript. This table now includes patient demographics and clinical profiles from healthy patients, before and after treatment with RCI-1502.
We sincerely hope that we have addressed the comments to the satisfaction of the reviewers.
Sincerely,
Dr Olaia Martínez-Iglesias
Department of Medical Epigenetics
EuroEspes Biomedical Research Center
Bergondo, 15165
Corunna, Spain
E-mail: epigenetica@euroespes.com
Round 3
Reviewer 3 Report
The authors have responded sufficiently to the comments raised
Author Response
Thank you for your comments